# Social Frailty among Community-Dwelling Older Adults during the COVID-19 Pandemic in Korea: A Cross-Sectional Study

**DOI:** 10.3390/ijerph191911963

**Published:** 2022-09-22

**Authors:** Seunghye Choi, Hana Ko

**Affiliations:** College of Nursing, Gachon University, Incheon 21936, Korea

**Keywords:** social frailty, health conditions, intrinsic capacity, digital literacy, aged

## Abstract

Social frailty among older adults has become a growing concern from a public health perspective in the context of the coronavirus disease 2019 (COVID-19) pandemic. This study’s aim was to investigate the influence of various aspects of social frailty in community-dwelling older adults during the COVID-19 pandemic. This study carried out a secondary analysis of data collected from the 2020 National Survey of Older Koreans and performed multinomial logistic regression analysis to identify the predictive factors of social frailty. The affected factors for the social frailty group were health conditions (depression), behavioral and metabolic risk factors (exercise, nutritional status, current smoking status, drinking frequency), intrinsic capacity (cognitive functions, activities of daily living), and digital literacy (use of smartphone or tablet PCs). Since multidimensional factors could affect older adults’ social frailty, comprehensive strategies are urgently needed to reduce their rate of social frailty.

## 1. Introduction

High-quality social connections are essential for mental and physical health and well-being at all ages [1]. To enhance older adults’ life satisfaction, the importance of improving their physical and mental health, as well as their functional competence and continuous social relationships, has been emphasized. Therefore, the World Health Organization has highlighted the concept of “aging in place”, and emphasized the importance of social and cultural measures to support it [2]. The concept refers to aging-friendly means of establishing relationships with others and maintaining a functional state as much as possible in the place where one is staying [2]. Considering that older adults must increasingly rely on their (informal) social relationships and social environment, it is important that interventions for the prevention or delay of social frailty target all the relevant aspects for every individual [3].

Social frailty is defined as a continuum of being at risk of losing, or having lost, resources that are important for fulfilling one or more basic social needs during one’s life span [3]. In addition, social frailty should be explored taking into account contextual, social, and cultural considerations [4]. Social frailty increases the risks of cardiovascular disease, stroke, diabetes, cognitive decline, dementia, depression, anxiety, and suicide [1,5]. It also shortens lives, causes disabilities, and reduces the quality of life [1,6]. Meanwhile, this relationship, namely that social frailty leads to decrease health conditions, may be a reciprocal rather than a one-way relationship [7]. A previous study reported that physical frailty predicts the development of social frailty [7]. However, there have been few reports of the prevalence of social frailty and how it relates to older South Koreans’ health conditions.

The coronavirus disease 2019 (COVID-19) that has spread worldwide since December 2019 necessitated various guidelines, including quarantine and social distancing, to be applied around the world [8]. Hence, social frailty among older people, a growing public health and public policy concern, has been made even more salient by the COVID-19 pandemic [1,9]. For survival in a situation where non-face-to-face communication becomes common, an essential ability is digital literacy, which supports knowledge, skills, and attitudes about communication and participation in society and culture [10,11]. In 2017, older adults’ level of digital literacy was quite low, and it had a significant effect on their social activity participation and satisfaction [10]. Therefore, it is necessary to examine digital literacy changes during the pandemic situation and how they affect social frailty.

The 2020 National Survey of Older Koreans [12], being the first survey of the baby boomer generation in the period of older adulthood, is meaningful for reflecting on the situation of various older adults, especially since it was conducted during the COVID-19 pandemic. Therefore, this study’s aim was to investigate the influence of various aspects on the social frailty of community-dwelling older adults during the COVID-19 pandemic. In other words, this study identifies the differences in social frailty in older adults according to health conditions, behavioral and metabolic risk factors, intrinsic capacity, economic status, and digital literacy. Through this, it intends to provide basic data for establishing strategies to respond to various changes and prevent social frailty among older adults.

## 2. Materials and Methods

### 2.1. Study Design

This study was a secondary analysis of the data from the 2020 National Survey of Older Koreans [12] to identify the factors influencing older adults’ social frailty. Its conceptual framework was developed based on a literature review (Figure 1).

### 2.2. Data Collection and Participants

This study analyzed data obtained from the 2020 National Survey of Older Koreans, conducted every three years by the Korean Ministry of Health and Welfare. The 2020 original survey data was collected by 169 trained interviewers using the Tablet-PC Assisted Personal Interview (TAPI) method from 14 September to 20 November 2020. This study was conducted via face-to-face interview. Due to the COVID-19 situation, the investigators received prior training on the double quarantine guidelines related to the prevention of COVID-19. In addition, prior to the actual interview, a preliminary survey was conducted again on the status of quarantine of the participants against the corona virus [12]. The target population comprised older adults (65 years or older) who lived in communities in 17 cities and provinces across the country. The 2020 National Survey of Older Koreans sample was selected using a proportional two-stage stratified sampling method. First, the population was stratified by 17 metropolitan cities and provinces across Korea, and then, they were stratified again by neighborhoods in the nine provinces (but not in the seven metropolitan cities). The Ministry of Health and Welfare research team applied various weights in the raw data to ensure the accuracy of estimations. The weight of the raw data was adjusted by considering the weights for households and individuals [12]. Of the 10,097 respondents, 9920 were selected, while 167 were excluded since they comprised proxy respondents and 10 owing to missing responses. The researchers received coded data, which were used for secondary data analysis.

### 2.3. Measures

#### 2.3.1. Social Frailty

Social frailty, a multidimensional concept, is classifiable into five categories [13]: (1) going out (not participating in any leisure and social activities, such as travel, hobbies, learning or studying, social clubs, networking, political and social groups, volunteering, senior citizen centers, community centers for older adults); (2) visiting friends (no); (3) feeling worthless (yes); (4) living alone (yes); and (5) contact with someone (not communicating with relatives including siblings, friends, neighbors, and acquaintances by phone, text message, or e-mail). Respondents with none, one, and two or more of these components were classified into the robust, social prefrailty, and social frailty groups, respectively [5,13,14].

#### 2.3.2. Health Conditions

Health conditions included the number of diagnosed chronic diseases, prescribed medicines, and depression.

Depression was measured using the Korean version of the Short Form Geriatric Depression Scale (K-SGDS) [15], which contains a total of 15 items. Each item was allocated 1 and 0 points for responses of “yes” and “no,” respectively, and the total score ranged from 0 to 15 points. The scores were classified as normal, suspected mild depression, and severe depression if less than 5 points, between 6 to 9 points, and more than 10 points, respectively [15]. The Cronbach’s alpha of K-SGDS was 0.89 at the time of its development, and 0.85 in this study.

#### 2.3.3. Behavioral and Metabolic Risk Factors

Behavioral and metabolic risk factors included exercise, nutritional status, current smoking status, drinking frequency during the past year, alcohol consumption, and body mass index (BMI) (kg/m^2^). Moreover, alcohol drinkers were divided into: (1) low-risk drinkers (consumption of less than 3 units of alcohol per day for men, and less than 2 units per day for women) and (2) risk drinkers according to the low-risk drinking guidelines in Korea [16].

Nutritional status in older adults was measured using “Determine Your Nutritional Health”, a tool developed by the Nutrition Screening Initiative [17]. This instrument consists of 10 items, each of which is rated from 1 to 4, with possible scores ranging from 0 to 21. Accordingly, nutritional status was categorized as: good (0 to 2 points), moderate risk (3 to 5 points), and high risk (6 points or more). In this study, respondents with 3 points or more was classified into the nutritional risk group.

#### 2.3.4. Intrinsic Capacity

Intrinsic capacity included cognitive functions, muscle power (sitting and standing up 5 times), fall experiences, Activities of Daily Living (ADL), and Instrumental Activities of Daily Living (IADL).

Cognitive functions were measured using the 19-item Mini-Mental State Examination for Dementia Screening (MMSE-DS) tool [18], having a maximum score of 30 points, with higher scores being indicative of higher cognitive functions. This tool has been standardized by age, gender, and educational levels for normative cognitive function assessment in older adults in South Korea [18]. Its Cronbach’s alpha was 0.82 in a previous study and 0.90 in this study.

#### 2.3.5. Economic Status

Economic status included annual income and job presence.

#### 2.3.6. Digital Literacy

Digital literacy was measured using a 3-item questionnaire: “Do you use Smartphones or tablet PCs? (yes/no),” “Do you use computers? (yes/no),” and “Do you utilize electronic devices for information retrieval? (yes/no).”

### 2.4. Ethical Considerations

The 2020 National Survey of Older Koreans was approved by Statistics Korea (Approval No. 117071), and institution review board approval for the research was received prior to carrying out the survey from the Korea Institute for Health and Social Affairs vide IRB No. 2020-36. All participants provided written informed consent prior to participation and were informed that they could withdraw their consent at any time without any disadvantage. After obtaining approval for our study, we received raw data without personal identification information. Additionally, the study was approved by the Institutional Review Board of G University, to which the corresponding author was affiliated (IRB No. 1044396-202109-HR-189-01). All methods were performed in accordance with the relevant guidelines and regulations.

### 2.5. Data Analyses

The collected data were analyzed using the SPSS Win 22.0 program (SPSS, IBM Corp., Armonk, NY, USA), with the two-tailed significance level set at 0.05. To ascertain differences in the social frailty subgroups—robust, social prefrailty, and social frailty—a descriptive statistical analysis was performed on the measured variables using the chi-squared test and one-way ANOVA test (with Scheffé post hoc test). Further, multinomial logistic regression analysis was performed to identify the predictive factors of social frailty.

## 3. Results

### 3.1. Prevalence Rate of Social Frailty

Of the 9920 participants, 1218 (12.3%), 3612 (36.4%), and 5090 (51.3%) were classified into the robust, social prefrailty, and social frailty groups, respectively (Table 1), and the mean ages in these three groups were 72.1, 73.2, and 73.9 years, respectively (Table 1).

### 3.2. Differences in Health Conditions, Behavioral and Metabolic Risk Factors, and Intrinsic Capacity of Participants in the Robust, Social Prefrailty, and Social Frailty Groups

Those in the social prefrailty and social frailty groups were older than those in the robust group (*p* < 0.001). The robust group had more males than the social prefrailty or social frailty groups (*p* < 0.001). Education levels were higher in the robust group than in the other groups (*p* < 0.001), whereas the number of diagnosed chronic diseases or prescribed medicines was higher in the social prefrailty and social frailty groups than in the robust group (*p* < 0.001). Those in the robust group had lower depression levels and exercised more than the other groups (*p* < 0.001 and *p* < 0.001, respectively).

The frequency of exercise per week was significantly higher in the robust group than in the other groups (*p* < 0.001). While the robust group’s nutritional status was better than that of the social prefrailty and social frailty groups, there was a significant difference between both these groups’ nutritional status (*p* < 0.001). The number of current smokers was higher in the robust group than in the other groups (*p* < 0.001). Although drinking frequency in the past year was higher in the robust group than in the other groups (*p* < 0.001), alcohol consumption among alcohol drinkers was higher in the social frailty group than in the social prefrailty group (*p* = 0.015). The social frailty group had a lower BMI than the social prefrailty group (*p* = 0.003).

While the robust group’s cognitive function scores were higher than the other groups (*p* < 0.001), there was a significant difference in cognitive scores between the social prefrailty and social frailty groups. Muscle power was the highest and fall experiences were the lowest in the robust group, as compared to the other groups (*p* < 0.001 and *p* < 0.001, respectively). ADL and IADL were better in the robust and social prefrailty groups than in the social frailty group (*p* < 0.001) (Table 1).

### 3.3. Differences in Economic Status and Digital Literacy of Participants in the Robust, Social Prefrailty, and Social Frailty Groups

The robust group had more elderly with jobs than the social prefrailty or social frailty groups (*p* < 0.001), and the robust group’s annual income was higher than the other groups (*p* < 0.001). The older adults in the robust group used smartphones or tablet PCs, computers, or electronic devices more than the social prefrailty or social frailty groups (*p* < 0.001, *p* < 0.001, and *p* < 0.001, respectively) (Table 2).

### 3.4. Results of Multinomial Logistic Regression

We conducted multinomial logistic regression to identify the associated factors with the robust, social prefrailty, and social frailty groups, given a set of independent variables that included health conditions, behavioral and metabolic risk factors, intrinsic capacities, economic status, and digital literacy. Before performing the multinomial logistic regression, we tested the goodness of fit. Our model could be considered reasonably good since, with respect to Pearson and Deviance procedures, the significant values in our data were 0.925 and 1.00, i.e., *p* > 0.05. Moreover, the significance level of the likelihood ratio test for model-fitting information was *p* < 0.001, and the pseudo-R-square value was 0.115 (Table 3).

Depression (OR = 0.880, *p*< 001) and ADL dependency (OR = 0.847, *p* = 0.041) were lower in the robust group than in the social frailty group, while exercise (OR = 1.179, *p* = 0.016), good nutritional status (OR = 4.185, *p* < 0.001), moderate nutritional status (OR = 1.842, *p* = 0.006), current smoking (OR = 1.470, *p* < 0.001), frequency of alcohol drinking in the past one year (OR = 1.054, *p* = 0.011), cognitive functions (OR = 1.041, *p* < 0.001), and use of smartphones or tablet PCs (OR = 1.236, *p* = 0.008) were higher in the robust group than in the social frailty group (Table 3). Conversely, the social prefrailty group’s scores in depression (OR = 0.906, *p*< 001) and ADL dependency (OR = 0.930, *p* = 0.008) were lower in the social prefrailty group than in the social frailty group, while exercise (OR = 1.266, *p* < 0.001), good nutritional status (OR = 2.181, *p* < 0.001), moderate nutritional status (OR = 1.620, *p* < 0.001), and cognitive functions (OR = 1.011, *p* = 0.020) were higher in the social prefrailty group than in the social frailty group (Table 3).

## 4. Discussion

Based on a legal mandate, the National Survey of Older Koreans has been conducted every three years since 2008; this study analyzed data from the 2020 survey. The robust, social prefrailty, and social frailty groups comprised 12.3%, 36.4%, and 51.3% respondents, respectively, but a comparative analysis using the same method with the 2017 dataset, which comprised 12.8%, 42.5%, and 44.7% respondents, respectively, revealed that the prefrailty group’s ratio had decreased, while that of the social frailty group had increased [5]. While the mean age of those in the social prefrailty group was similar (73.9 years in 2017 versus 73.2 years in 2020), the mean age of the social frailty group decreased (75.6 years in 2017 versus 73.9 years in 2020) [5]. In particular, since the 2020 National Survey of Older Koreans was conducted in the context of COVID-19, the increase in the social frailty group of Korean older adults and the decrease in their average age could be related to this period effect. A previous study had also reported that a higher incidence of social frailty could be expected as a result of the COVID-19 pandemic [10]. The increasing social frailty among older adults might not be a temporary phenomenon given that social frailty has a direct correlation with neuropsychiatric disorders, such as depression, and the effects could last for a long time [10,14].

In this study, the affecting factors for the robust group when compared to the social frailty group were health conditions (depression), behavioral and metabolic risk factors (exercise, nutritional status, current smoking, drinking frequency), intrinsic capacity (cognitive functions, ADL), and digital literacy (using smartphones or tablet PCs). The affecting factors for the social prefrailty group when compared to the social frailty group were depression, exercise, nutritional status, cognitive function, and ADL. However, the values were not significant for smoking, alcohol intake, or use of smartphones or tablet PCs. According to a previous study, the mediators of social frailty were depression, physical deficiency, smoking, and cognitive impairment [19]. This study being a cross-sectional survey, it is difficult to determine the long-term effects of smoking and drinking. In fact, the robust group was found to smoke and drink more than the social frailty group, even after adjusting for age and sex. Moreover, the robust group had a higher drinking frequency over the past year than the other groups. This result was consistent with the results of a previous longitudinal study, which reported that the alcohol intake group had a lower incidence of functional limitations than the abstention groups [20]. However, the mean amount of alcohol consumption among alcohol drinkers was higher in the social frailty group than in the social prefrailty group. Moreover, the number of risk drinkers was higher in the social frailty group than in the other groups. Although light to moderate alcohol intake has been reported to prevent all-cause deaths and cardiovascular diseases in older adults [21], chronic heavy drinking is known to be associated with a number of serious neurological conditions, such as dementia or cognitive decline [22]. Therefore, it is necessary to manage frail older adults’ alcohol intake for ensuring that that they do not consume excessive amounts. Although in this study, the robust group smoked more than the social frailty group, smoking is known to be associated with the development or worsening of frailty in the general population [23]. Social frailty leads not only to a deterioration in the quality of life among older adults, but also to various health problems later in life [24]. Therefore, even in robust older adults, regulation of smoking and drinking is needed for health management in later years.

In this study, the robust and social prefrailty groups had lower depression, more exercise, and better cognitive functions than the social frailty group. These results were consistent with those of a previous study [25]. Social frailty has been defined as the lack of social resources, social activities, and self-management abilities that are important for fulfilling basic social needs [3]. Indeed, social activity is important because older adults with fewer social activities tend to have more cognitive decline [26]. Moreover, socially isolated older adults were less physically active and had poorer nutritional behaviors, eventually leading to health problem [27]. Depressive symptoms were significantly associated with social media usage, social support, and intergenerational relationships [28]. Hence, depressive symptoms in socially frail older adults needs to be addressed, and exercising at home could help these individuals to maintain their mental health during the COVID-19 pandemic [29]. Therefore, during the pandemic when social distancing is required, it is necessary to have interventions such as exercise, ICT (information and communication technology), including social media at home, that can reduce depression while maintaining social relationships.

On the one hand, various factors were found to affect the health status of the social frailty group, as compared to the other groups. On the other hand, since December 2019, as the world has been affected by the COVID-19 pandemic, direct interpersonal relationships have become difficult for most people. Although all age groups are at risk of contracting COVID-19, older adults are more vulnerable and thus have a greater need for social restraint and social distancing [30]. In the era of COVID-19, possessing digital literacy is very important for obtaining quality information quickly, reducing loneliness, and maintaining interpersonal relationships [31]. The robust group was found to have better digital literacy than the social prefrailty group. In fact, according to this study’s results, only 51.2% of the older adults responded that they used smartphones or tablet PCs. Hence, there is an urgent need to establish a digital infrastructure for vulnerable older adults.

This study had some limitations. First, only the variables included in the raw data were used in the secondary data set. Second, although the respondents’ representativeness was established by recruiting them nationwide and using a well-designed sampling method, there was a limit to explaining the causality of results, since this was a cross-sectional study.

## 5. Conclusions

In conclusion, social frailty in older adults in Korea has increased during the COVID-19 pandemic. There were differences in the variables between the robust and social prefrailty groups, as compared to the social frailty group. The affecting factors for the robust group were health conditions (depression), behavioral and metabolic risk factors (exercise, nutritional status, current smoking, drinking frequency), intrinsic capacity (cognitive functions, ADL), and digital literacy (use of smartphones or tablet PCs). Comprehensive strategies to improve health conditions, control behavioral and metabolic risk factors, improve intrinsic capacity, and digital literacy are urgently needed to reduce the number of older adults experiencing social frailty.

## Figures and Tables

**Figure 1 ijerph-19-11963-f001:**
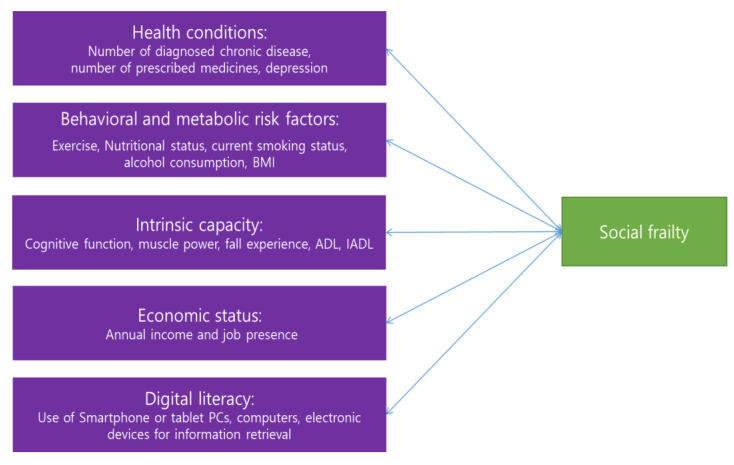
Conceptual framework for this study.

**Table 1 ijerph-19-11963-t001:** Differences in health conditions, behavioral and metabolic risk factors, and intrinsic capacity of participants in the robust, social prefrailty, and social frailty groups (N = 9920).

Characteristics	Category	Total (n = 9920, 100%) Mean (SD)	Robust ^a^ (n = 1218, 12.3%) Mean (SD) or n (%)	Social Prefrailty ^b^ (n = 3612, 36.4%) Mean (SD) or n (%)	Social Frailty ^c^ (n = 5090, 51.3%) Mean (SD) or n (%)	χ^2^/F	*p*	Scheffé Test
**Demographic characteristics**								
Age (years)		73.4 (6.5)	72.1 (5.8)	73.2 (6.2)	73.93 (6.9)	41.89	<0.001	a < b, a < c
Male		3971 (40.0)	604 (49.6)	1467 (40.6)	1900 (37.3)	62.36	<0.001	
Education (years)		8.2 (4.0)	9.11 (3.6)	8.38 (3.9)	7.84 (4.1)	55.67	<0.001	a > b, a > c
**Health conditions**								
Number of diagnosed chronic diseases		1.8 (1.5)	1.5 (1.3)	1.7 (1.4)	2.01 (1.6)	87.17	<0.001	a < b, a < c
Number of prescribed medicines		1.8 (1.5)	1.4 (1.2)	1.6 (1.4)	1.95 (1.7)	80.67	<0.001	a < b, a < c
Depression		3.4 (3.4)	2.2 (2.3)	2.7 (2.7)	4.17 (3.8)	317.51	<0.001	a < b, a < c
**Behavioral and metabolic risk factors**								
Exercise (yes)			692 (56.8)	2041 (56.5)	2454 (48.2)	69.66	<0.001	
Exercise (weekly frequency)		4.8 (1.8)	5.2 (1.6)	4.8 (1.8)	4.6 (1.8)	29.81	<0.001	a > b, a > c
Nutritional status	Mean score	2.0 (3.0)	0.9 (1.77)	1.5 (2.3)	2.6 (3.5)	251.30	<0.001	a < b, a < c, b < c
	Good	7058 (71.1)	1068 (87.7)	2807 (77.7)	3183 (62.5)	486.07	<0.001	
	Moderate nutritional risk	1943 (19.6)	123 (10.1)	625 (17.3)	1195 (23.5)			
	High nutritional risk	919 (9.3)	27 (2.2)	180 (5.0)	712 (14.0)			
Current smoking (yes)		1089 (11.0)	206 (16.9)	375 (10.4)	508 (10.0)	50.40	<0.001	
Drinking frequency over the past year		1.2 (1.8)	1.5 (1.90)	1.1 (1.7)	1.1 (1.7)	27.79	<0.001	a > b, a > c
Risk drinkers		1943 (52.8)	248 (45.4)	672 (51.1)	1023 (56.3)	22.32	<0.001	
Alcohol consumption		3.9 (2.3)	3.8 (2.31)	3.7 (2.1)	4.0 (2.3)	4.22	0.015	b < c
BMI	Mean score	23.6 (2.6)	23.6 (2.36)	23.7 (2.5)	23.5 (2.7)	4.03	0.018	b > c
	Underweight	213 (2.2)	14 (1.2)	67 (1.9)	132 (2.6)	15.71	0.003	
	Normal	7208 (72.9)	917 (75.6)	2605 (72.3)	3686 (72.6)			
	Obesity	2468 (25.0)	282 (23.2)	929 (25.8)	1257 (24.8)			
**Intrinsic capacity**								
Cognitive scale		24.3 (5.3)	25.8 (4.04)	24.7 (4.9)	23.7 (5.7)	90.42	<0.001	a > b, a > c, b > c
Muscle power	Performed	7285 (73.4)	1009 (82.8)	2754 (76.2)	3522 (69.2)	159.64	<0.001	
Fall experiences	Yes	633 (6.4)	49 (4.0)	195 (5.4)	389 (7.6)	30.73	<0.001	
ADL		7.2 (1.1)	7.0 (0.3)	7.1 (0.7)	7.3 (1.4)	45.58	<0.001	a < c, b < c
IADL		10.6 (2.6)	10.1 (0.9)	10.3 (1.8)	10.9 (3.2)	78.52	<0.001	a < c, b < c

Note: BMI = body mass index; ADL = activities of daily living; IADL = instrumental activities of daily living.

**Table 2 ijerph-19-11963-t002:** Differences in economic status and digital literacy of participants in the robust, social prefrailty, and social frailty groups (N = 9920).

Characteristic		Total (n = 9920, 100%) Mean (SD) or n (%)	Robust ^a^ (n = 1218, 12.3%) Mean (SD) or n (%)	Social Prefrailty ^b^ (n = 3612, 36.4%) Mean (SD) or n (%)	Social Frailty ^c^ (n = 5090, 51.3%) Mean (SD) or n (%)	χ^2^/F	*p*	Scheffé Test
**Economic status**								
Job	Yes		516 (42.4)	1320 (36.5)	1778 (34.9)	23.48	<0.001	
Annual income (10,000 Won)		2700.6 (3976.7)	3256.5 (2777.8)	2698.8 (2500.4)	2568.8 (4944.8)	14.74	<0.001	a > b, a > c
**Digital literacy**								
Use of smartphones or tablet PCs	Yes	5083 (51.2)	766 (62.9)	1939 (53.7)	2378 (46.7)	116.43	<0.001	
Use of computers	Yes	543 (5.5)	97 (8.0)	239 (6.6)	207 (4.1)	43.19	<0.001	
Utilization of electronic devices (information retrieval)		3955 (45.6)	599 (52.8)	1509 (47.0)	1847 (42.8)	39.96	<0.001	

Note: PC = personal computer.

**Table 3 ijerph-19-11963-t003:** Results of multinomial logistic regression.

	Robust versus Social Frailty					Social Prefrailty versus Social Frailty				
	*B*	*p*	Exp (B)	95% CI		B	*p*	Exp (B)	95% CI	
				Lower	Upper				Lower	Upper
(Constant)	−2.913	0.001				−1.364	0.005			
Age	0.004	0.548	1.004	0.991	1.017	0.006	0.148	1.006	0.998	1.015
Gender (male)	0.116	0.135	1.123	0.965	1.307	0.019	0.721	1.019	0.919	1.130
Number of diagnosed chronic diseases	−0.036	0.209	0.965	0.912	1.020	−0.015	0.413	0.985	0.951	1.021
Depression	−0.128	0.000	0.880	0.856	0.903	−0.098	0.000	0.906	0.892	0.921
Exercise (yes)	0.165	0.016	1.179	1.032	1.348	0.236	0.000	1.266	1.156	1.386
Nutritional status (Good)	1.432	0.000	4.185	2.783	6.293	0.780	0.000	2.181	1.809	2.630
Nutritional status (Moderate)	0.611	0.006	1.842	1.193	2.846	0.483	0.000	1.620	1.332	1.971
Current smoking (yes)	0.385	0.000	1.470	1.198	1.804	0.039	0.633	1.040	0.887	1.219
Drinking frequency during the last one year	0.053	0.011	1.054	1.012	1.098	−0.003	0.866	0.997	0.968	1.027
BMI	0.008	0.561	1.008	0.981	1.036	0.017	0.060	1.017	0.999	1.035
Cognitive scale	0.040	0.000	1.041	1.025	1.057	0.011	0.020	1.011	1.002	1.020
Fall experiences during the last one year (yes)	−0.166	0.310	0.847	0.615	1.167	−0.042	0.665	0.959	0.794	1.159
ADL	−0.158	0.041	0.854	0.734	0.994	−0.073	0.008	0.930	0.881	0.982
Annual income	0.000	0.570	1.000	1.000	1.000	0.000	0.046	1.000	1.000	1.000
Use of smartphones or tablet PCs (yes)	0.212	0.008	1.236	1.056	1.448	0.085	0.111	1.089	0.981	1.210
Likelihood ratio test chi^2^	1017.889									
*p*	<0.001									
Pseudo R^2^ (Nagelkerke)	0.115									

Note: CI = confidence interval; BMI = body mass index; ADL = activities of daily living; PC = personal computer.

## Data Availability

The authors have no authority over the data, and the data is provided upon request to the Ministry of Health and Welfare.

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
