# Peer review of "Social Frailty among Community-Dwelling Older Adults during the COVID-19 Pandemic in Korea: A Cross-Sectional Study"

_ijerph, 2022, doi:10.3390/ijerph191911963_

Round 1
Reviewer 1 Report
The topic of the article on social fragility and analyzing it in the context of the pandemic is interesting and the results found contribute to the knowledge to achieve healthy aging.
Authors are advised to:
• Do a review of the writing of the article.
• In the results section where Table 3 is described in lines from 199 to 208, it is recommended to interpret the results of the multinomial logistic regression and not only include the OR data and the p values.
Author Response
We appreciate the time and effort spent by the reviewers to provide feedback and helpful suggestions for improving the article. We carefully considered the reviewers’ comments and made numerous edits to the manuscript and highlights. We responded to each of the reviewers’ comments and outlined my manuscript revisions in the included Response to Reviewers document. We are hopeful that the revised manuscript will be worthy of publication in International Journal of Environmental Research and Public Health. We thank the editor and the reviewers for the thoughtful comments and suggestions. Our responses and a summary of changes made to the manuscript are listed below.

Reviewer 2 Report
I think this is a great article about social frailty based on a 2020 national survey of older adults in Korea. The article is definitely worth publishing. There are a couple clarifications needed prior to publication.
1. I believe the authors are describing that they did secondary data analysis of data available from a national survey. Please clarify in the methods section if the methods were secondary data analysis.
2. in section 2.2 of the manuscript, It is not clear to the reader if the authors actually participated in the creation and administration of the national survey or if the authors are simply describing the methods of the national survey team.
3.. Figure 1: for the conceptual framework, the figure layout makes it seem like the 5 boxes (health conditions through digital literacy) all precede/lead to social frailty. I think you are trying to show there is simply a relationship, correct? Perhaps the authors could add bidirectional arrows to the lines connecting the boxes to social frailty to show that the relationship probably goes both ways? (e.g., <---->)
4. Table 1. For ease of reading, I would recommend rounding to the nearest 10th after the decimal (i.e. for age instead of 73.44 (6.53) change it to 73.4 (6.5)). Similarly, for X2 could it be displayed as 41.9 instead of 41.886?
5. Page 1 Line 30, the statement that older adults must increasingly rely on informal social relationships "due to policy measures aimed at reducing the financing of formal care and support". Is it as simple as that one reason? I would imagine it is more complicated than that. For example, in America it seems maybe more people are relying on their informal supports longer because the idea of a nursing home is even worse now during the pandemic than it was before. I think the authors could take out the portion of the sentence "--due to policy measures aimed at reducing the financing of formal care and support--" and the sentence would be good without that statement.
6. Page 1, line39. I would recommend deleting the word "temporal" and simply state "this relationship."
Author Response

(The authors gave the same response as above.)

Reviewer 3 Report
Review comments for “Social Frailty among Community-dwelling Older Adults during the COVID-19 Pandemic in Korea: A Cross Sectional Study.”
This paper investigates the influence of various factors on social frailty of community-dwelling older adults during COVID-19 pandemic. The findings of the paper may have potential policy implications that may point to the importance of increasing digital literacy of frail older adults while expanding digital infrastructural for all. Below, I would like to concentrate on issues that need attention for authors in their process of revision.
First of all, the construct of “social frailty,” was introduced by citing prior literature. The validity of this construct needs proper discussion. For instance, in the cultural context of Korea the concept was not adequately discussed. What does “social contact” mean? Does it include children’s or family members’ visits? In Asian cultures, this is a very important component of “contacts.” While older adults may not go out to visit others; others, such as adult children or family members may come in to visit them. These visits, as well as phone calls others made to the older adults constitute social contacts. The 5 measures of social frailty (p. 3, lines 82-86) did not appear to reflect social contacts “received.” Spelling out these activities, especially in the social and cultural contexts of Korea, will be helpful for readers. Otherwise, research instruments used in the west sound vague or abstract in the context of Korea.
Secondly, the study sample included older adults in 17 cities and provinces in Korea. Did these sampling sites comprise urban and rural areas? Social isolation among residents in rural areas is generally greater than those in urban settings. Social frailty rural residents experience may also differ from urban older adults. This is an important issue and deserve attention. If the national sample did not include any rural residents, I would question the level of its representativeness.
Thirdly, this study used “Tablet-PC assisted personal interview (TAPI) method (p. 3, line 74-75). Were these interviews face-to-face? Or were they telephone interviews? Since it was during Sept to Nov. 2020, while Covid-19 was still very prevalent, I assume the “personal” face-to-face interviews were difficult, if not unlikely. If it was telephone interviews, how did researchers handle hearing issues. It is well known that older adults have difficulty hearing clearly, even when interviews were conducted face-to-face. Different interview methods may yield different quality of study results. Clarify your study methods is important.
Fourthly, how are the sample study subjects identified? What is your process of randomization? Or is it random? How did you handle the issue of cognitive disability among older adults? Was there an instrument of screening to identify cognitive disability levels of study subjects. I assume people in their 70s (which is your mean age) are quite likely to have a certain percentage who suffer from dementia or Alzheimer’s disease. This issue needs to be addressed.
Among your study instruments, you included 19-item “Cognitive functions” (p.4, line 116-118). If study subjects scored low in their cognitive function, are they capable of participating the study about social contacts and social frailty? Does it make sense to include them in the study?
Furthermore, you stated in the findings section that “the robust group had more males…” and “Education levels were higher in the robust group” (Lines 159-160). Did you run an interaction term between gender and education?
Overall, I have reservations about the study methodology and sampling process.
I enjoyed reading the paper. Covid-19, to a certain extent, highlighted the urgent needs for digital literacy among all, particularly older adults, whether in Korea or other parts of the world; Covid-19 also heightened the demand and importance for digital communication to maintain social well-being for all, especially older adults. This paper adds to the literature with this argument.
Author Response

(The authors gave the same response as above.)
